# A Cross-Sectional Analysis of Food Perceptions, Food Preferences, Diet Quality, and Health in a Food Desert Campus

**DOI:** 10.3390/nu14245215

**Published:** 2022-12-07

**Authors:** Kaitlyn J. Aldaz, Sigry Ortiz Flores, Rudy M. Ortiz, L. Karina Diaz Rios, Jaapna Dhillon

**Affiliations:** 1Department of Molecular & Cell Biology, School of Natural Sciences, University of California, Merced, CA 95343, USA; 2Division of Agriculture and Natural Resources, University of California, Merced, CA 95343, USA; 3Department of Nutrition and Exercise Physiology, University of Missouri, Columbia, MO 65203, USA

**Keywords:** nutrition, food environment, barriers, sensory perception, college, diet formation

## Abstract

The sensory properties of foods guide food choices and intake, importantly determining nutritional and health status. In communities that have inconsistent access to nutritious foods, such as food deserts, food taste perceptions and preferences have yet to be explored. The purpose of this study was to examine how taster status (supertaster vs. non-taster) and food security status (high or marginal vs. low or very low) influences food taste intensities, food preferences and perceptions, and diet quality in a cohort of students from a food desert campus in the Central Valley of California. Moreover, the complex relationship of socioeconomic status, race/ethnicity, and sex on cardiometabolic and cognitive health warrants further examination. Two hundred fifty participants (aged 18–24 years) living in a food desert campus were recruited in 2018 for this cross-sectional study where participants underwent taste tests on selected fruits, vegetables, and nuts, and clinical tests (anthropometrics, blood glucose, blood pressure, and endothelial function), cognitive function tests (memory and attention), diet quality assessment (Healthy Eating Index (HEI)), and food preference and perception assessments. Food taste intensities were influenced by sex with bitter and umami taste intensities of several foods being perceived more intensely by males. Moreover, food liking was largely influenced by ethnicity with Hispanics having higher liking ratings for several foods compared with non-Hispanics. Both, Hispanics and females, had higher total fruit HEI scores and lower attention scores than non-Hispanics and males, respectively. Females also had lower blood pressure, reactive hyperemia index, and fasting blood glucose. Food-insecure individuals rated cost and convenience as more important factors for overall food consumption and had lower attention scores than those with higher food-security status. Future research should consider the complex interactions of factors such as taste and flavor perception, sex, ethnicity, prior exposure to foods, and other environmental factors when studying food preferences and health in young adults.

## 1. Introduction

The sensory properties of foods guide food choices and intake, making them key determinants of nutritional health [1]. Taste can impact food liking, help in anticipation of nutritional content of food, and collectively with other sensory properties, influencing satiation [2]. While salt, sweet, and umami (savory) tastes promote food liking, higher taste intensities and duration are associated with lower energy intake [2]. Conversely, foods that taste sour or bitter are not major determinants of energy intake [1]. Genetic differences in taste perception and sensitivity may influence food preferences [3]. For example, individuals with an allelic variation within the Taste receptor 2 member 38 (TAS2R38) gene perceive intense bitterness to thiourea compounds such as phenylthiocarbamide (PTC) or 6-n-propylthiouracil (PROP). Limited research suggests that such individuals (identified as supertasters) have lower preference for cruciferous vegetables, spinach, asparagus, and lower overall vegetable intake [4,5,6,7,8]. Associations of bitter taste perception with health outcomes have also been observed. For example, higher bitter taste perception is associated with heightened innate immune protection [9], and in older adults with metabolic syndrome, lower diabetes risk [10].

In communities that have inconsistent access to nutritious foods, such as food deserts, the sensory perceptions of and preferences for fruit, vegetables, and nuts have yet to be explored. This research study was conducted on a food desert university campus [11], in an area that is ranked as the 3rd highest for child food-insecurity in the nation when compared to all counties with over 10,000 hungry children [12]. Limited food variety and high cost are the most salient perceived structural barriers to obtaining nutritious food on a food desert campus [11]. Disparities in access to food during early transitional years can have detrimental health effects over time as children may develop maladaptive eating behaviors [13,14,15]. Other factors such as ethnicity, sex, and socioeconomic status (SES) can also influence taste and food preferences, and food choices [16,17]. Individuals identifying as Hispanics and African Americans are found to rate various taste intensities (sweet, salty, sour, and bitter) higher than non-Hispanics with more notable differences being apparent in males than females [16]. Since sensory properties appear to determine food choices and intake [2], these could potentially play a role in the diet-related disparities among racial and ethnic minority groups [18]. In addition, there are documented differences in food choices by gender. Women consistently consume more vegetables, fruits, dietary fiber, and less fats, mainly due to greater nutritional knowledge and weight-control motivation, and stronger health beliefs compared to men [17,19,20,21,22]. Moreover, socioeconomic indicators such as education, income, and occupation are often intertwined, and can collectively influence food choices and diet quality [23]. Individuals with a high SES are more likely to have a nutritious diet [23]. Conversely, those with low SES consume fewer fruits and vegetables due to financial constraints [23,24], and more energy-dense foods, hence, their diets tend to be inconsistent with the dietary guidelines [23,25]. Sub-optimal dietary patterns may detrimentally impact cardiometabolic [26] and cognitive health [27].

Various indicators in young adulthood can influence the formation of dietary patterns [23] and subsequently health [28]. Hence, examining how food perceptions and preferences varies by genetic taster status, SES, ethnicity, and sex and the extent to which such variability may explain disparities in cardiometabolic and cognitive health among people living in food deserts is important. One objective of this study was to examine how taster status (supertaster vs. non-taster) and food security status (high or marginal vs. low or very low) influences food taste intensities and liking, food preferences, and diet quality in a cohort of students from a food desert campus. The second objective was to assess the effects of ethnicity, sex, and perception of financial stability on the aforementioned factors. The third objective was to examine the effects of taster and food security status, sex, ethnicity, and socioeconomic factors on cardiometabolic and cognitive outcomes.

## 2. Methods

### 2.1. Study Participants

Participants were university students recruited from a university campus in Central California in 2018 using convenience sampling. The study was advertised around the college campus using flyers, classroom presentations, and word-of-mouth. Interested participants were sent a screening questionnaire via email to check the inclusion and exclusion criteria. The inclusion criteria included being a healthy young adult (age 18–24 years): (a) with no food allergies, (b) not taking medications that affect metabolism and appetite, (c) willing to comply with the study protocol, and (d) nonsmoker (>1 year or more). The exclusion criteria included current use of: (a) illicit drugs, (b) medications that affect metabolism or appetite, (c) anti-inflammatories, analgesics, and antibiotics, (d) drug therapies targeting diabetes or pre-diabetes, hypertension, cardiovascular disease, or dyslipidemia, and/or gastrointestinal disease and/or bariatric surgery. Participants provided informed consent and the study was approved by the University’s Institutional Review Board (IRB).

### 2.2. Study Design

The cross-sectional study was conducted over two visits. In the first visit participants’ clinical and dietary outcomes were measured. During the second visit, the participants underwent sensory and cognitive tests. They were required to fast for 8–12 h prior to both visits. The safety of participants in this IRB-approved study was ensured though the use of proper techniques for data collection. Researchers underwent training to ensure that any potential risks to participants were minimized. Participants’ data were kept confidential by not including identifiable information on any data file. The file linking the participant’s study ID to their details was stored separately on an offline encrypted drive that was password-protected.

### 2.3. Participant Categorizations

To determine taster status, participants were asked to sample and note the taste quality and intensity of paper strips coated with phenylthiocarbamide (PTC, 3–5 µg/strip) and a control strip (no PTC) [29,30,31]. Participants were categorized as supertasters if they perceived the PTC strip as extremely bitter, or as non-tasters if they perceived the PTC strip as tasteless or any other taste in comparison to the control strip. Food security status was assessed using the 6-item US household food security module [32], which was administered online via Qualtrics survey platform (Qualtrics, Provo, UT, USA). Participant’s perception of financial stability was also recorded by asking if they considered themselves financially stable.

### 2.4. Sensory and Dietary Assessments

Sensory tests were performed on participants in a group format. Using 9-point hedonic and general labeled magnitude scales [33], participants were asked to describe the flavor, and rate the liking and taste intensities (salt, sweet, sour, umami, and bitter) of 18 widely available fruits, nuts, and vegetables: almonds, asparagus, avocado, broccoli, carrots, celery, clingstone peach, collard greens, green beans, kiwi, olives, peanuts, pistachios, plums, pomegranates, and strawberries, using 9-point hedonic and general labelled magnitude scales, respectively [33]. Participants also rated the importance of several factors such as availability, convenience, cost, health, nutrition, and taste on influencing consumption of the aforementioned foods in general on a scale of 1 (not important) to 9 (extremely important). In addition, paired preference tests [34,35,36] were conducted for similar foods matched on nutritive value but differing in cost.

Participants dietary information was collected using single time-point Automated Self-Administered (ASA) 24 h food recalls [37] and Healthy Eating Index (HEI) scores were calculate [38,39].

### 2.5. Clinical Outcomes

All measurements were collected by trained researchers and all participants were consented to participate prior to any collections. The anthropometric measures pertained to adiposity, body mass, body mass index (BMI), height, waist circumference, and waist-to-hip ratio. Hip, thigh, and waist circumferences were measured using standard techniques with a measuring tape. Height was measured using a wall-mounted stadiometer. Body mass (kg) and body composition were measured using a calibrated bio-electrical impedance scale (Model BC-418, Tanita Inc., Arlington Heights, IL, USA). The biochemical measures were comprised of fasting blood glucose using a glucometer (Model HM100005). Trained researchers used lancets to prick the side of a finger. A drop of blood was placed on the glucometer test strips until the surface was fully covered. The strip was then inserted into the glucometer to calculate the blood glucose concentrations of participants. Systolic and diastolic blood pressure were measured using an automated blood pressure device (Model HEM 780, Omron Corporation, Kyoto, Japan). The blood pressure cuff was placed on the non-dominant arm of participants. Participants were instructed to not talk while the blood pressure measurements were collected. All measurements were taken twice and averaged to obtain a final value. Endothelial function of a subset (*n* = 100) of participants was assessed using an EndoPat device (Itamar Medical Ltd., Caesarea, Israel).

### 2.6. Cognitive Outcomes

Cognitive tests for attention and memory were also conducted in a group format, 15 min after the sensory tests. In the immediate memory test [40], participants were read a list of ten words and after 2 min were asked to recall and write the words on to a piece of provided paper. The number of correct responses was recorded. In the attention test, participants were given a recording blank with 4 min to cross out the letter d accompanied with two dashes in different combinations on a sheet with fourteen lines of 47 letters comprised of either d or p with one, two, three or four dashes [40]. The errors of omission (missing target d’s) and errors of commission (crossing wrong targets) were recorded and total error percentage (E%) computed by the researchers. Quantitative performance (the total number of items processed, TN), qualitative performance (the total number of items processed minus total errors, TNE) and concentration performance (the total number of correct items marked minus errors of commission, CP) were also recorded.

### 2.7. Statistical Analysis

Analyses comprised testing effects of taster status (supertaster vs. non-taster), food security status (high or marginal vs. low or very low), perception of financial stability (yes vs. no), sex, and ethnicity (Hispanic vs. non-Hispanic) on food taste and flavor intensities and liking ratings. Sensory data such as food taste and flavor intensities and liking and importance ratings were analyzed for the aforementioned factors using Wilcoxon rank sum test. Non-parametric effect sizes were calculated using rank-biserial correlation. The quantiles of the sensory test data are presented in Appendix A. Clinical, cognitive function, and dietary variables were analyzed for the aforementioned factors using Welch’s *t*-test. *P*-values were false discovery rate (FDR) corrected and Q < 0.1 was considered statistically significant. FDR correction for variables were done among similar variables (i.e., separately for clinical, cognitive, dietary, and rating variables) and for the sensory data across foods. Chi-square goodness of fit test was used to determine which food was preferred in the paired set for the preference tests, and whether participants could correctly identify the foods in the sensory tests. The effects of aforementioned factors on the paired food preference data and the food identification data were analyzed using chi-square test of independence. FDR correction was applied across the paired sets for the preference test.

## 3. Results

Two hundred fifty participants (174 female and 76 male) participated in the cross-sectional study. Participants were categorized based on sex (female vs. male), ethnicity (Hispanics vs. non-Hispanics), taster status (supertasters vs. non-tasters), food security status (high or marginal vs. low or very low), and participants’ perception of financial stability (yes vs. no).

### 3.1. Food Identification

The food identification results are shown in (Table 1). The majority of participants correctly identified most foods except for collard greens, which was correctly identified by only 4% of the participants. Even in an unblinded test, 60% of the participants were not familiar with collard greens. 

### 3.2. Taste Intensities and Food Liking Ratings 

#### 3.2.1. Taster Status Had Minimal Effect on Taste Intensities and Liking Ratings of Foods

Supertasters perceived collard greens as more bitter (Effect size: 0.19) than non-tasters (*p* < 0.05). However, non-tasters perceived strawberries as more bitter (0.20), and peanuts (0.17) as saltier and as having higher flavor intensity (0.19) than supertasters (*p* < 0.05). Non-tasters also had higher liking ratings for olives (0.15) than supertasters (*p* < 0.05). After FDR correction, no statistically significant differences were observed for taste intensities and liking ratings by taster status.

#### 3.2.2. Food Security Status and Financial Stability Perception Influenced the Taste Intensities and Liking Ratings of Specific Foods 

Participants with low food security status perceived celery as saltier (0.20) than those with high food security status (Q < 0.1). Participants who perceived themselves as financially stable had higher liking ratings for olives (0.27) than those with low food security status, respectively (Q < 0.1).

#### 3.2.3. Ethnicity and Sex Influenced the Flavor and Taste Intensities, and Liking Ratings of Specific Foods

Participants who identified as Hispanic perceived carrots (0.20) and plums (0.21) to be of higher flavor and avocado to be less fatty (0.23) than non-Hispanics (0.38) (Q < 0.1). This group also had higher liking ratings for avocado (0.16), plums (0.27), carrots (0.23), almonds (0.22), broccoli (0.18), cauliflower (0.18), peaches (0.17) but lower ratings for olives (0.24) than non-Hispanics (Q < 0.1).

Males perceived pomegranate as sweeter (0.25), and almonds (0.18), avocado (0.18), peanuts (0.23), and pistachio (0.17) as more bitter, and almonds (0.22), avocado (0.34), kiwi (0.17), and plums (0.19) as more umami than females (Q < 0.1). Females had higher liking ratings for strawberries (0.24) and peach (0.23) than males (Q < 0.1).

### 3.3. Preference Test Results 

The paired preference test results indicate that a significant majority preferred carrots (89%) over celery (11%), spinach (84%) over collard greens (16%), avocado (82%) over olives (18%), pistachios (77%) over peanuts (23%), and green beans (64%) over asparagus (36%) (*p* < 0.05). Differences in preference for collard greens vs. spinach and avocado vs. olives were found by ethnicity (Q < 0.1). More non-Hispanics (67%) preferred collard greens compared to Hispanics (33%) while more Hispanics preferred spinach (59%) compared to non-Hispanics (41%). More Hispanics (59%) preferred avocado compared to non-Hispanics (41%) while more non-Hispanics preferred olives (63%) compared to Hispanics (37%). 

Differences in preference for broccoli vs. cauliflower and peach vs. plum were found by sex as well (Q < 0.1). Males preferred broccoli (70%) over cauliflower (30%), while females were approximately equally divided, i.e., broccoli (49%) vs. cauliflower (51%). Females preferred peach (63%) over plums (37%) and males preferred plums (57%) over peach (43%). 

### 3.4. Importance Ratings of Taste, Cost, Availability, Convenience, and Nutrition for Food Consumption

#### 3.4.1. Perception of Financial Stability and Food Security Status Influenced Perceptions around Food

Participants’ who perceived themselves as not financially stable rated the cost of fruits (0.46), vegetables (0.40), and nuts/seeds (0.34), and food (0.29) in general, and availability of fruits (0.19) and nuts/seeds (0.21), and convenience of food (0.21), nuts/seeds (0.16), and vegetables (0.16) as more important factors for consumption, and rated nutrition (0.19) as less important and perceived greater lack of control over selection and preparation of food (0.29) than those who perceived themselves as financially stable (Q < 0.1, Figure 1). Participants’ who perceived themselves as not financially stable also had higher concern for hypertension (0.16) and obesity (0.17) than those who perceived themselves as financially stable (Q < 0.1).

More specifically, in the sensory tests, participants’ who perceived themselves as not financially stable rated the importance of convenience for broccoli (0.21), peaches (0.25), spinach (0.22), strawberries (0.23), and kiwi (0.17), and the importance of cost for strawberries (0.27), almonds (0.21), carrots (0.22), peaches (0.21), pomegranates (0.22), spinach (0.21), broccoli (0.20), green beans (0.19), kiwi (0.19), avocados (0.18), plums (0.18), and cauliflower (0.16) higher than those who perceived themselves as financially stable (Q < 0.1).

Participants of low food security status rated importance of cost for fruits (0.29), vegetables (0.27), and foods in general (0.23), and importance of convenience for nuts/seeds (0.27) and foods in general (0.20) as higher than those of high food security status (Q < 0.1, Figure 2). In the sensory tests, participants with low food security status perceived the cost of avocados (0.29), almonds (0.16), broccoli (0.17), carrots (0.17), celery (0.16), strawberries (0.16), and green beans (0.20) as more important than those with a high food security status (Q < 0.1).

#### 3.4.2. Sex Influenced the Importance of Availability and Taste for Consumption of Specific Foods

Females rated the importance of availability for consumption of strawberries (0.32), and taste for consumption of broccoli (0.29), green beans (0.30), spinach (0.26), almonds (0.19), avocado (0.17), carrots (0.20), and cauliflower (0.18) more than males (Q < 0.1).

#### 3.4.3. Taster Status Influenced the Importance of Nutrition and Health Benefits for Consumption of Specific Foods

Supertasters rated the nutrition of cauliflower (0.23) and perceived health benefits of cauliflower (0.21) and strawberries (0.21) as more important factors for consumption than non-tasters (Q < 0.1).

### 3.5. Clinical, Cognitive Function, and Dietary Outcomes

In general, females had higher body and trunk fat% and heart rate normalized augmentation index but lower systolic and diastolic blood pressure, mean arterial pressure, reactive hyperemia index, fasting blood glucose, and waist circumference than males (Q < 0.1, Table 2). Hispanics had higher body fat% than non-Hispanics (Q < 0.1, Table 2). Participants with high food security status had higher CP and lower E% than those with low food security status (Q < 0.1, Table 2). Moreover, non-Hispanics had higher CP and TNE, and lower E% scores than Hispanics (Q < 0.1, Table 2). Males had higher CP, TN, and TNE scores than females (Q < 0.1, Table 2).

Healthy eating (HEI) scores differed by ethnicity and sex. Hispanics had higher total fruit and sodium HEI scores than non-Hispanics (Q < 0.1, Table 2). Moreover, females had higher total, total fruit, whole fruit, and sodium HEI scores but lower dairy and protein HEI scores than males (Q < 0.1, Table 2).

## 4. Discussion

The current study suggests that taster status had minimal effect on taste or flavor intensities and liking ratings of foods in this cohort of young adults attending a food-desert campus. Although it is widely believed that genetic taste sensitivity to bitter compounds may influence preferences and consumption of foods with bitter undertones such as cruciferous vegetables [5], and subsequently health outcomes, our study does not provide conclusive evidence of the same. Biological factors interact with culture and environmental factors to determine eating choices and dietary patterns [11,41,42]. Thus, taste intensity perception used as an indicator of food preferences must be interpreted in the context of such interaction [43]. Moreover, exposure to foods in utero, during breastfeeding, and in early childhood, can influence preferences and consumption later in life [43,44,45,46]. The more familiar a stimulus (for example, food) becomes with repeated exposure [47], the greater the likelihood of an individual preferring and consuming that food [43]. Hence, consuming bitter vegetables consistently during childhood may condition supertasters to prefer them as adults regardless of bitter intensity perception.

Sex influenced the taste intensities and liking rating of several foods in the sensory tests as well as HEI scores in our study. While males had higher taste intensities for several foods, females rated taste as a more important factor for consumption of foods, especially vegetables, compared to males. In contrast, Caucasian women of European descent demonstrated that perceived taste sensations such as sour, bitter, and salty in solutions made from standard tastants was more intense than their male counterparts; however, the association between taste intensity and taste preference was not well-defined [48]. Other studies suggested that there is possibly an interaction of taster genotype with sex on taste perception [49]. Hence, food preferences are likely determined by a complex interaction of taste, sex, and possibly race/ethnicity determining prior exposure to foods. Females also had higher HEI total score and total fruit, whole fruit, and sodium scores but lower total dairy and total protein scores than males. When compared to men, women are more conscientious of health factors that contribute to appearance, which may result in healthier dietary choices [50]. Evidence also suggests that men are less aware of dietary recommendations than women [51,52]. Women more frequently purchase and prepare food and therefore have a better understanding of the nutritional value of foods than men [53]. Healthier dietary patterns are associated with better cardiometabolic outcomes [54,55,56,57]. Importantly, in our study, higher total HEI scores in females paralleled better blood pressure, waist circumference, and fasting blood glucose compared to males.

Ethnicity influenced flavor intensities of selected foods but not food taste intensities in the sensory tests in the present study. Other studies demonstrate that when responsiveness to basic tastants were assessed, participants identifying as Hispanics were more responsive to taste stimuli compared to non-Hispanics [16]. Ethnicity can serve as an indicator of future dietary preferences as commonly consumed foods during childhood may influence food choices [58,59,60]. In our study, participants who identified as Hispanics had higher liking ratings for selected fruits, vegetables, and nuts, as well as higher HEI total fruits and sodium scores than those who identified as non-Hispanics. This is supported by a recent systematic review that demonstrates that Hispanics have higher fruit intakes than non-Hispanic Whites [61]. Analysis of 24 h recalls from a cohort of the Hispanic Community Health Study reports that the sodium intake among Hispanics is much higher than recommended intake [62]. These differences may become less pronounced with acculturation, as Hispanics who become socialized to American culture may drift away from traditional foods and adopt dietary intake patterns that are similar to non-Hispanic Whites [63,64].

In our study, food cost and convenience, particularly of fruits and vegetables, were salient factors influencing consumption behaviors among food insecure and financially unstable participants. Low SES and food insecurity status are obstacles for purchasing healthy foods, thereby influencing food consumption [65]. The relatively high cost of fruits and vegetables serves as a deterrent to their consumption. The lower energy density and limited shelf-life of fresh fruits and vegetables may lead to the selection and purchase of more energy-dense and shelf-stable cheaper foods. This is especially the case among those of low SES, for whom financial instability compels food procurement patterns that favors the consumption of shelf-stable vs. nutritious food [66]. The convenience of accessing, preparing, and transporting foods is an important factor in determining the frequency of consumption [11,67]. Foods that are easier to carry, such as prepackaged foods, are more commonly selected over foods that take more effort to prepare [67]. Consumption of fruits and vegetables often requires washing, peeling, and/or cooking, all of which are time-consuming. In our study, only those who perceived themselves as financially unstable perceived a greater lack of control over the selection and preparation of food, rated nutrition as less important for food consumption, and had higher concern for hypertension and obesity. Food-insecure and low-income people have an increased burden of diet-related chronic diseases due to limited access to healthy foods, higher stress, and an unfavorable built environment [28,68].

In our cohort of young adults, participants with high food-security status had better cognitive function outcomes compared with those with low food-security status. Healthier dietary patterns are associated with better cognitive function outcomes particularly in older adults [69,70]. Food insecurity may exacerbate the decline in cognitive function with aging [71]. Our findings are congruent with previous studies in children that have found a correlation between SES and d2 task performance [72]. Low SES children scored much lower on the d2 task performance compared to higher SES children [72] suggesting that food security and SES are interlinked [73]. Areas heavily populated with families of lower SES frequently and typically have restricted access to a variety of recreational and learning materials that may be seen in more affluent areas, impacting early cognitive development [74]. SES components such as family income, and parental education and occupation may have a complex effect on early cognitive development [74], which could impact cognitive performance later in adulthood [75,76,77].

Cognitive function outcomes differed by ethnicity in our study with non-Hispanics having better attention outcomes compared with Hispanics. Previous research has mostly focused on older adults and there appears to be an effect of timing of migration on cognitive function [78]. US-born Hispanics show lower cognitive function when compared with US-born non-Hispanic Whites and foreign-born Hispanics [78]. Adverse socioeconomic environments can impact US-based Hispanics’ cognitive health through limited access to educational and occupational opportunities, psychosocial stress, and systemic racism [79]. Additionally, studies have shown US-based Hispanics are at higher risk of developing hypertension, obesity, and T2D, all of which can impair cognitive health [80]. 

Cognitive function differences were also observed based on sex with males having better attention outcomes, but similar memory outcomes compared with females in the present study. The existing evidence regarding cognitive differences by sex is inconclusive, owing to the influence of several biological and psychosocial factors [81]. Hormonal differences resulting from epigenetics changes during developmental stages of life, such as adolescence, may contribute to brain function and cognitive differences [81,82,83]. In addition, environmental factors such as culture, gender roles, and school demands can impact sleep, thereby impacting cognitive performance [84].

The study findings should be interpreted based on the study’s strengths and limitations. An important strength of this study lies in the novelty of examining effects of socioeconomic, ethnicity, and sex characteristics on food perceptions and preferences of people living in food deserts. Other strengths pertain to the large sample size, and use of rigorous sensory, cardiometabolic, and cognitive measures. Limitations pertain to the inherent restrictions of a single time-point cross-sectional design that does not allow for repeated measures and causal inference [85]. To expand generalizability, future research should explore food perceptions and preferences in other age groups, across the lifespan, among other ethnicities, and under different socioeconomic and geographical contexts [86]. 

## 5. Conclusions

In a cohort of young adults living in a food desert campus, food taste intensities were largely influenced by sex, and food liking by ethnicity. Food-insecure individuals reported cost and convenience as more important factors for food consumption and had lower cognitive function (attention) scores than those with higher food security status. In addition, there were observations of associations of sex and ethnicity with diet quality as well as cognitive and health outcomes. Although these findings may partly be driven by physiological factors, these are likely more significantly impacted by sociocultural and built environmental factors and systemic inequities [41,42]. Future research should consider the complex interactions of diverse factors such as taste and flavor perception, sex, ethnicity, prior exposure to foods, and other socioeconomic and environmental factors for assessment of food preferences and health.

## Figures and Tables

**Figure 1 nutrients-14-05215-f001:**
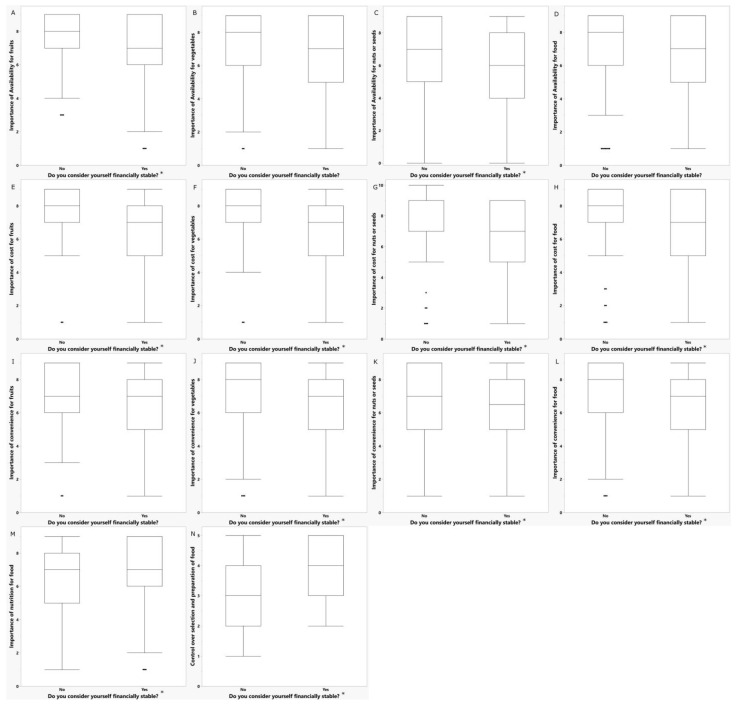
Ratings of importance of factors for food consumption by perception of financial stability. * Q < 0.1 for comparisons within categories: (**A**) Importance of availability for fruit; (**B**) Importance of availability for vegetables; (**C**) Importance of availability for nuts or seeds; (**D**) Importance of availability for food; (**E**) Importance of cost for fruits; (**F**) Importance of cost for vegetables; (**G**) Importance of cost for nuts or seeds; (**H**) Importance of cost for food; (**I**) Importance of convenience for fruits; (**J**) Importance of convenience for vegetables; (**K**) Importance of convenience for nuts or seeds; (**L**) Importance of convenience for food; (**M**) Importance of nutrition for food; (**N**) Control over selection and preparation of food.

**Figure 2 nutrients-14-05215-f002:**
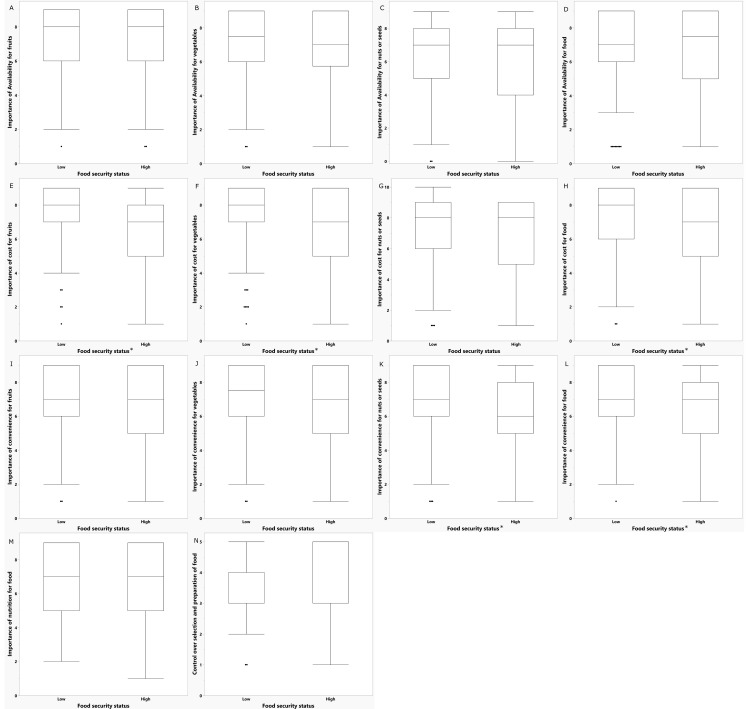
Ratings of importance of factors for food consumption by food security status. * Q < 0.1 for comparisons within categories: (**A**) Importance of availability for fruit; (**B**) Importance of availability for vegetables; (**C**) Importance of availability for nuts or seeds; (**D**) Importance of availability for food; (**E**) Importance of cost for fruits; (**F**) Importance of cost for vegetables; (**G**) Importance of cost for nuts or seeds; (**H**) Importance of cost for food; (**I**) Importance of convenience for fruits; (**J**) Importance of convenience for vegetables; (**K**) Importance of convenience for nuts or seeds; (**L**) Importance of convenience for food; (**M**) Importance of nutrition for food; (**N**) Control over selection and preparation of food.

**Table 1 nutrients-14-05215-t001:** Food identification results in name-blinded sensory tests.

Food Type	Correctly Identified	Incorrectly Identified
Almonds	98%	2%
Asparagus	83.60%	59%
Avocado	99.60%	0.40%
Broccoli	99.60%	0.40%
Carrots	100%	0%
Cauliflower	96.80%	3.20%
Celery	95.60%	4.40%
Collard Greens	4.40%	95.60%
Green Beans	63.20%	8%
Kiwi	99.20%	0.80%
Olives	97.60%	2.40%
Peach	72.80%	32%
Peanuts	92%	8%
Pistachio	92.80%	7.20%
Plums	76.80%	23.20%
Pomegranate	90.09%	9.91%
Spinach	76%	24%
Strawberries	100%	0%

**Table 2 nutrients-14-05215-t002:** Clinical, cognitive function, and HEI characteristics by taster, food security status, perception of financial stability, sex, and ethnicity.

	Taster Status		Food Security Status		Perception of Financial Stability		Sex		Ethnicity	
	Super Tasters (*n* = 169)	Non-Tasters (*n* = 81)	High Food Security Status (*n* = 102)	Low Food Security Status (*n* = 138)	Financially Stable (*n* = 127)	Financially Unstable (*n* = 113)	Males (*n* = 76)	Females (*n* = 174)	Hispanic (*n* = 136)	Non-Hispanic (*n* = 112)
Clinical Outcomes										
Age (year)	20.35 ± 1.4 ^1^	20.16 ± 1.56	20.25 ± 1.47	20.31 ± 1.46	20.17 ± 1.36	20.42 ± 1.56	20.3 ± 1.43	20.28 ± 1.46	20.36 ± 1.39	20.21 ± 1.51
Body Mass (kg)	71 ± 18	73 ± 21	71 ± 18	71 ± 20	71 ± 17	71 ± 21	79 ± 20 *	68 ± 18	72 ± 19	71 ± 19
BMI (kg/m^2^)	26 ± 6	26 ± 7	26 ± 7	26 ± 6	26 ± 6	27 ± 7	26 ± 5	26 ± 7	27 ± 6	26 ± 7
Waist Circumference (cm)	83 ± 13	84 ± 15	83 ± 13	83 ± 14	82 ± 12	84 ± 15	87 ± 13 *	82 ± 14	85 ± 13	81 ± 14
Total Fat %	30 ± 10	29 ± 11	29 ± 10	30 ± 11	28 ± 10	31 ± 11	20 ± 7 *	33 ± 9	*31 ± 10 **	27 ± 11
Trunk Fat %	28 ± 10	27 ± 11	27 ± 10	27 ± 11	27 ± 10	28 ± 11	22 ± 9 *	30 ± 10	29 ± 10	26 ± 11
Systolic BP (mmHg)	110 ± 13	112 ± 12	112 ± 12	109 ± 12	112 ± 12	109 ± 12	122 ± 12 *	106 ± 9	110 ± 13	112 ± 12
Diastolic BP (mmHg)	72 ± 8	72 ± 7	72 ± 7	72 ± 7	72 ± 7	72 ± 8	74 ± 7 *	71 ± 7	72 ± 8	72 ± 7
Mean Arterial Pressure	85 ± 8	86 ± 8	85 ± 8	84 ± 8	85 ± 8	84 ± 8	90 ± 8 *	83 ± 7	84 ± 9	86 ± 8
FBG (mg/dL)	91 ± 12	90 ± 11	90 ± 10	91 ± 13	90 ± 10	92 ± 13	93 ± 15 *	90 ± 10	91 ± 11	91 ± 13
Reactive Hyperemia Index	1.74 ± 0.56	1.64 ± 0.43	1.65 ± 0.43	1.76 ± 0.59	1.78 ± 0.52	1.64 ± 0.54	1.87 ± 0.61 *	1.62 ± 0.46	1.74 ± 0.57	1.66 ± 0.46
Augmentation Index	−6.6 ± 12.07	−6.12 ± 10.81	−5.3 ± 16.21	−7.07 ± 7.63	−5.95 ± 14.35	−6.8 ± 8.43	−8.97 ± 6.01	−5.33 ± 13.23	−7.5 ± 7.88	−5.01 ± 15.27
Augmentation Index@75	−9.14 ± 12.09	−6.26 ± 10.11	−7.49 ± 16.32	−8.97 ± 7.02	−9.01 ± 14.82	−7.68 ± 6.88	−12.12 ± 8.26 *	−6.47 ± 12.33	−10.35 ± 7.8	−5.3 ± 14.75
Cognitive Function Outcomes										
Total Number of Correct Words Recalled	5.8 ± 1.4	5.7 ± 1.4	5.9 ± 1.4	5.6 ± 1.4	5.7 ± 1.3	5.8 ± 1.5	5.6 ± 1.3	5.8 ± 1.5	5.6 ± 1.4	5.9 ± 1.5
Concentration Performance (CP)	153.13 ± 32.76	144.47 ± 32.04	156.87 ± 30.21 *	146.53 ± 33.89	154.83 ± 28.99	146.53 ± 36.08	158.03 ± 31.86 *	146.96 ± 32.61	145.74 ± 30.75 *	155.88 ± 34.3
Quantitative Performance (TN)	361.58 ± 67.27	354.17 ± 69.78	366.71 ± 65.16	355.94 ± 70.11	365.96 ± 64.21	354.4 ± 72.06	373.76 ± 68.32 *	352.81 ± 67.13	352.08 ± 59.86	368.1 ± 76.02
Qualitative Performance (TNE)	350.34 ± 67.45	338.28 ± 67.68	356.91 ± 64.59	340.9 ± 69.24	354.76 ± 62.46	339.77 ± 72.46	362.2 ± 67.98 *	339.55 ± 66.5	337.74 ± 59.93 *	357.12 ± 74.75
Errors of Omission	9.29 ± 12.34	13.43 ± 16.57	8.01 ± 10.08 *	12.62 ± 16.35	9.31 ± 10.75	12.17 ± 17.19	9.79 ± 11.41	11 ± 14.95	11.99 ± 16.53	9.15 ± 9.92
Errors of Commission	1.95 ± 3.19	2.46 ± 4.72	1.78 ± 4.11	2.43 ± 3.56	1.88 ± 4.09	2.46 ± 3.46	1.78 ± 4.71	2.26 ± 3.25	2.35 ± 4.79	1.83 ± 1.84
Total Error % (E%)	3.17 ± 3.73	4.44 ± 4.43	2.69 ± 2.78 *	4.23 ± 4.67	3.04 ± 2.96	4.17 ± 4.94	3.12 ± 3.13	3.78 ± 4.33	4.07 ± 4.87 *	3.03 ± 2.53
HEI Scores										
HEI Total	49.2 ± 14.2	51.4 ± 14.0	51.5 ± 14.9	48.6 ± 13.5	51.6 ± 13.9	47.8 ± 14.2	47.2 ± 12.6 *	51.1 ± 14.7	49.9 ± 14.0	50.0 ± 14.5
HEI Total Fruits	1.9 ± 2.1	1.8 ± 2.0	2.1 ± 2.1	1.7 ± 2.0	2.0 ± 2.1	1.7 ± 2.0	1.1 ± 1.7 *	2.2 ± 2.1	2.2 ± 2.1 *	1.4 ± 1.9
HEI Whole Fruits	2.1 ± 2.3	2.0 ± 2.3	2.4 ± 2.4	1.8 ± 2.3	2.3 ± 2.3	1.9 ± 2.3	1.2 ± 2.0 *	2.5 ± 2.3	2.4 ± 2.4	1.7 ± 2.2
HEI Total Vegetables	3.0 ± 1.8	3.4 ± 1.8	3.1 ± 1.8	3.2 ± 1.8	3.2 ± 1.8	3.1 ± 1.8	3.1 ± 1.7	3.2 ± 1.8	3.0 ± 1.8	3.4 ± 1.7
HEI Total Dairy	4.3 ± 3.7	4.7 ± 3.8	4.3 ± 3.7	4.5 ± 3.7	4.6 ± 3.7	4.2 ± 3.7	5.2 ± 3.9 *	4.1 ± 3.6	4.4 ± 3.6	4.4 ± 3.8
HEI Protein	4.0 ± 1.6	4.3 ± 1.4	4.1 ± 1.6	4.1 ± 1.5	4.2 ± 1.5	4.0 ± 1.6	4.5 ± 1.2 *	4.0 ± 1.6	4.1 ± 1.6	4.2 ± 1.5
HEI Sodium	3.6 ± 3.5	3.6 ± 3.7	4.0 ± 3.6	3.3 ± 3.5	3.8 ± 3.6	3.4 ± 3.5	2.9 ± 3.4 *	3.9 ± 3.6	4.2 ± 3.5 *	2.9 ± 3.5

^1^ All such values are Mean ± SD; BMI = body mass index; FBG = fasting blood glucose; * Q < 0.1 for comparisons within categories.

## Data Availability

Data will be available on reasonable request.

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
