# Peer review of "A Cross-Sectional Analysis of Food Perceptions, Food Preferences, Diet Quality, and Health in a Food Desert Campus"

_nutrients, 2022, doi:10.3390/nu14245215_

Round 1

Reviewer 1 Report

The work deals with an interesting issue, however, before publication, it needs to be improved.

Abstract:

- No purpose of the study.

- Usually, abbreviations are not used in the abstract. Introduction:

Line 43: TAS2R38 - please explain the abbreviation

line 93: IRB - please explain the abbreviation.

Lines 79-82 - These two sentences should be removed to the methodological section.

There is no justification for conducting the research in the introduction.

Information about food security in this region is worth adding.

The purpose of the study is not precise and difficult to follow.

Methodology - When and among whom was the study conducted?

- What sampling was used? random, on purpose, others?

- How the selection process took place, it can be presented on the plot.

- The authors write that the study was conducted at the University Campus in central California; why? What was the basis for selecting the research facility?

- Please attach the questionnaire and the scales used as supplementary material.

- Subsection 2.5. No description of the methodology of the measurements carried out. The authors write: "standard technics", what does it mean? Please describe the measurement procedure in detail. Who took blood for the determination of blood glucose levels? Blood pressure measurement - who carried out the measurement and according to what procedure. Is the measuring device standardized?

- What procedures were used to ensure the safety of the study participants?

- How were the data from the study participants protected?

Results

Lines 168-171: the study group needed to be characterized.

Tables and figures in the text should be placed immediately after the reference.

Table 1, the headings need to be edited, perhaps abbreviations or font reduction.

Line 389: The authors indicate the strengths of the "large sample" study.

How was the study sampling calculated? Are the study participants a representative group of the study population?

Reviewer 2 Report

Thank you for the opportunity to read the text.
The abstract lacks information on when the research was conducted and where - (food deserts)?
The purpose of the study is not specifically stated.

Hypotheses or research questions should be stated along with the main objective at the end of the introduction. Such summaries of the importance of a given topic of interest.

Table 2 - please reduce the font - the text and data are unreadable in this form.

In my reflection, the discussion is not adequate.
What I find missing in the discussion is a reference to the research questions/hypotheses and a confrontation of the results obtained with the results of other researchers according to the questions/hypotheses posed.
The Conclusion section is missing the Limitations subsection.

I hope that my suggestions will help the authors to improve the article in the acceptance of the reader.

Round 2

Reviewer 1 Report

Accept in present form.

Reviewer 2 Report

Thank you for considering the suggestions I offered.
In truth, a few more things could be improved, but they are technical in nature and I think the editing team will bring them to the attention of the authors themselves.
I have no more constructive submissions for improvement.
I wish you good luck.